# Non-Destructive Estimation of Leaf Size and Shape Characteristics in Advanced Progenies of *Coffea arabica* L. from Intraspecific and Interspecific Crossing

**DOI:** 10.3390/plants14192985

**Published:** 2025-09-26

**Authors:** Carlos Andres Unigarro, Aquiles Enrique Darghan, Daniel Gerardo Cayón Salinas, Claudia Patricia Flórez-Ramos

**Affiliations:** 1Discipline of Plant Physiology, National Coffee Research Center, Cenicafé, Manizales 170009, Colombia; 2Departamento de Agronomía, Facultad de Ciencias Agrarias, Universidad Nacional de Colombia, Bogotá 111321, Colombia; aqedarghanco@unal.edu.co; 3Departamento de Ciencias Agrícolas, Facultad de Ciencias Agrarias, Universidad Nacional de Colombia, Palmira 763533, Colombia; dgcayons@unal.edu.co; 4Plant Breeding, National Coffee Research Center, Cenicafé, Manizales 170009, Colombia; claudia.florez@cafedecolombia.com

**Keywords:** hybrids with *Coffea canephora*, leaf length and width, allometric relationships, Montgomery model, principle of similarity

## Abstract

Non-destructive measurement of leaf size based on leaf length and/or width is a simple, economical, and precise methodology. Leaf morphometric indicators were measured on 55 coffee progenies obtained from intraspecific and interspecific crosses. The estimation of parameters in the models and the testing of hypotheses related to these were performed. The relationships between leaf width and length, the ellipticity index, and leaf size were subsequently analyzed with a partitioning algorithm. The groups were then compared using Hotelling’s T^2^ test. In coffee, the Montgomery model allowed for an adequate estimation of leaf size for each progeny, hybridization type, and grouped data. An α value of 0.67000 for the Montgomery model was consistent. This finding indicates that it is a suitable model for both individual and groups of progenies. The model based on the “principle of similarity” was found to be suitable only on a per-progeny basis. Certain characteristics, such as the leaf width-to-length ratio, ellipticity index, and leaf size, modify the parameter fit to inherent values. Similarly, leaves with a higher width-to-length ratio were the most elliptical for coffee, according to the groupings found. The estimation of coffee leaf size improves if the selected model considers whether they come from specific progenies or groups of progenies.

## 1. Introduction

Leaf size (L_S_) plays a significant role in the ecological strategies of plant species and their interactions with ecosystem function. It varies from 0.01 cm^2^ to 10,000 cm^2^ and is linked to plant growth and development in different environments. For example, many large-leaved species are prevalent in the tropics, whereas small-leaved species are found in deserts and at high elevations and latitudes [1,2,3,4]. This is attributed to the direct influence of L_S_ on radiation interception, self-shading, and leaf temperature, which affect leaf photosynthesis, transpiration, and respiration [5,6,7,8]. Methods for quantifying L_S_ are categorized into destructive and non-destructive approaches [9,10]. The former offer high precision and accuracy because they require dissecting the leaves and direct measurement using various instruments, including a planimeter [11], automatic leaf area meters [e.g., Delta-T Devices (Cambridge, UK), LI-COR (Lincoln, NE, USA)] [12], and, more recently, image recognition software [13]. Destructive methods have significant limitations, such as the inability to record the growth of the same leaf blade over time [14], the alteration of some physiological parameters after defoliation [9], and the high cost of time and resources when the work is intensive [15]. In contrast, non-destructive methods avoid excising leaves and estimating leaf area using allometric scaling models based on leaf length and width [16] and, more recently, use automatic recognition technologies on mobile platforms [12]. Non-destructive methods are relatively easy and quick to implement in situ and allow for temporal monitoring of the same leaf without major physiological disturbances to the plant but at the cost of reduced precision and accuracy compared with destructive methods [14]. Notably, non-destructive methods for estimating L_S_ require a prior calibration process involving destructive leaf measurements [10].

Allometric models estimate L_S_ based on a power law [L_S_ = α X^β^ + ɛ], where α is the normalization constant, β is the scaling exponent with a value between 1 and 3, and X can be the product of two morphometric measurements of the leaf, typically length (L) and width (W), although models can also be developed with a single dimension [17,18]. The models described below are written as statistical models, so an error term (ɛ) is included, for which a mean of 0 and variance $\sigma_{\varepsilon }^{2}$ are assumed. In this way, the “principle of similarity,” proposed by Thompson in 1917, posits that the area of an object is proportional to the square of its length [19]. This has often been demonstrated with exponents close to 2 [17], especially when estimated for leaves with simple shapes (e.g., corn leaves) [16]. Montgomery [20] determined a direct proportional relationship between L_S_ and the product of L and W (LW) for corn leaves using the model L_S_ = α LW + ɛ. The model proposed by Montgomery has been widely adopted and validated for calculating L_S_ in herbaceous plants, woody plants, and crops [9,16]. In the case of LW, it is defined as a rectangular area enclosing the leaf [L_S_ = LW] and overestimates L_S_. Therefore, the α parameter of the Montgomery model acts as a correction factor for size and an indicator of leaf shape [18]. The α parameter typically ranges between ½ (for triangular leaves) and π/4 (for elliptical leaves) [21], with α = 2/3 being a value used for L_S_ estimation in the case of ovate leaves [22]. However, this parameter can have values less than ½ in palmatisect leaves [e.g., *Cecropia obtusifolia*], pinnatisect leaves [e.g., *Lomatia silaifolia*], and fern leaves [e.g., *Davallia solida*] [18] or higher than π/4 in cordate leaves [e.g., *Ipomoea nil*] [9], obovate leaves [e.g., *Gleditsia triacanthos*] [23], and superelliptical leaves [e.g., *Quercus pannosa*] [24].

Leaf shape (S_h_) is determined by genetic [25] and environmental [26] factors and plays a functional role in the fitness, light interception, and evapotranspiration of plants [27,28,29]. This condition influences the trade-off between the support cost for the leaf area and its photosynthetic return [30,31]. The ratio between leaf width and length (R_WL_) has proven to be a good indicator of the geometric characteristics of S_h_ because of its positive correlation with the fractal dimension of the leaf, particularly for various broad-leaved plants and especially for ovate leaves [32]. When the median R_WL_ exceeds 0.125, it is negatively correlated with the scalar exponent of leaf dry mass vs. L_S_, which decreases toward 1. This finding indicates that as leaves become wider, dry mass increases proportionally with increasing leaf area; therefore, wide leaves have a lower support cost than narrow leaves do [31]. Another relevant indicator of S_h_ is the ellipticity index (E_I_), which was developed from the Montgomery model to quantify the deviation of ovoid leaves from a standard ellipse [21]. Unlike other parameters, such as sphericity and dissection indices, which evaluate deviation from a perfectly circular shape, E_I_ does not require measuring the leaf perimeter and achieves a better fit for elliptical, ovate, and obovate leaves, whose geometries differ significantly from those of a circular shape [24]. However, E_I_ does not differ between leaves with a narrower base than the apex (obovate leaves) or the inverse situation (ovate leaves).

Of the 124 species that constitute the genus *Coffea* (family: Rubiaceae), *Coffea arabica* L. stands out as the most economically important, with an estimated market value between 200 and 250 billion dollars annually and providing livelihoods for at least 60 million people worldwide [33]. *C. arabica* plants, according to their size, can be tall-statured, reaching a height of 4–6 m in a growth cycle (5–6 years from planting or pruning), or short-statured, with a height of 2–3 m [14,34]. Their growth is monopodial and continuous, resulting in the formation of a sequence of metamers, each composed of an internode with two leaves per node [35]. The coffee leaf is petiolate, opposite, simple, and attached to the margin and is of a mesophyll size (4500–18,225 mm^2^); it is not lobed and has an undented and sinuous margin type, an acuminate apex (drip tip), and a convex base that is slightly rounded, according to the criteria for classifying leaf characteristics established by Ellis et al. [36]. In this context, quantifying L_S_ is important because the sum of the area of individual leaves on a plant allows for the estimation of leaf area (L_A_), which has a direct effect on fruit production in coffee [37]. Therefore, methods that allow easy, economical, and precise L_S_ quantification are necessary [18].

In coffee, allometric models based on measuring L and W have proven to be effective indicators for determining L_S_; in addition, these models also allow for sequential, non-destructive measurements of leaves on an individual over time [10,14,15,38,39,40,41,42,43,44,45,46,47]. These characteristics have been highly useful in studies of leaf growth and development [48], radiation interception and distribution [49], source–sink relationships [50], water deficit and excess [51,52], and modeling [53] in coffee. The estimation of the α parameter using the Montgomery model in coffee leaves has yielded the following general values: (a) 0.6434 in 8 *C. arabica* genotypes located at three latitudes [44], (b) 0.6440 in *C. arabica* cv. Catuaí 144 Rojo [46], (c) 0.6670 in *C. arabica* cv. Bourbon Amarillo [38], (d) 0.6800 in 47 Ethiopian *C. arabica* genotypes [45], (e) 0.70125 in *C. arabica* (cv. Bourbon Amarillo, cv. Catuaí 2147, cv. Mundo Novo 2190, cv. Typica), *Coffea canephora* (cv. Conilon 513, cv. CC 3580), Timor Hybrid (interspecific cross between *C. arabica* and *C. canephora*), and the F2 421-4 segregating population (interspecific cross between *C. arabica* and *C. canephora*) [15], and (f) 0.7480 in *C. arabica* cv. Caturra under shade provided by *Erythrina poeppigiana* trees [41]. However, little research has been conducted on the association of the α parameter with L_S_ or S_h_ or on the differences between intraspecific and interspecific crossing in coffee. Given this, the present study aimed to evaluate L_S_ as a function of L and W using the Montgomery model for 55 coffee progenies derived from intraspecific and interspecific crossing. Additionally, the proportionality between L_S_ and the square of L was evaluated using a power-law model to determine whether the “principle of similarity” holds true for coffee leaves. Ultimately, the characteristics of S_h_ and L_S_ were examined to establish possible groupings of progenies originating from intraspecific and interspecific crossing.

## 2. Results

### 2.1. Fit of Models M1 and M2 for Estimating LS and Research Hypotheses

For Model 1 (M1: L_S_ = α LW + ɛ), the first hypothesis (H1) showed supporting evidence in nine progenies ({6}, {2}, {5}, {4}, {17}, {15}, {7}, {53}, and {50}) (9/26 = 34.6%) of the 26 progenies originating from intraspecific *C. arabica* hybridization crosses (Intra.H.), with an estimated α parameter for these progenies that was statistically greater than the α value of the grouped data (Pd), with values ranging between 0.67469 and 0.68592 (0.5% and 2.2%). Conversely, in five progenies ({1}, {12}, {10}, {8}, and {52}) (5/26 = 19.2%), the α parameter was statistically smaller, ranging between 0.65302 and 0.66686 (2.6% and 0.6%) (Table 1). Furthermore, in 10 progenies ({21}, {24}, {25}, {20}, {23}, {33}, {43}, {47}, {38}, and {29}) (10/29 = 34.4%) of the 29 progenies with interspecific hybridization cross group between *C. arabica* and *C. canephora* (Inter.H.), the α parameter was statistically greater than the α of Pd, ranging between 0.67517 and 0.68576 (0.6% and 2.2%). In contrast, 11 progenies ({30}, {27}, {39}, {31}, {35}, {42}, {36}, {48}, {44}, {45}, and {28}) (11/29 = 37.9%) had smaller α values than Pd did, ranging between 0.65397 and 0.66780 (2.5% and 0.4%) (Table 1). In 12 of the Intra.H. progenies (12/26 = 46.1%), no statistical evidence against H1 was found, whereas this occurred in only eight of the Inter.H. progenies (8/29 = 27.6%) (Table 1). This suggests that the α parameter of Pd can be used to adequately estimate L_S_ for more individual Intra.H. progenies than Inter.H. progenies. Additionally, across all the evaluated progenies, only 36.4% (20/55) showed an α parameter statistically equivalent to that of Pd. Notably, α showed statistical evidence favoring H1 for both the Intra.H. and Inter.H. types (Table 1). The minimum and maximum values for L, W, and L_S_ are presented in Appendix A.

The α parameters of M1, calculated for each progeny, hybridization type, and Pd, yielded confidence intervals that did not contain the lowest α value reported in coffee (α = 0.64340) [44], which provides statistical evidence supporting second hypothesis (H2). Similarly, these intervals did not contain the highest α value found in coffee (α = 0.70125) [15], which offers statistical evidence supporting third hypothesis (H3) (Table 1). All confidence intervals fell within the theoretical range (1/2, π/4) of the Montgomery model (M1) (Table 1).

With respect to the correction factor of 2/3 [18,22], the α parameter of M1 showed evidence supporting fourth hypothesis (H4) in 16 progenies ({6}, {2}, {5}, {4}, {14}, {17}, {3}, {15}, {16}, {7}, {11}, {53}, {54}, {55}, {50}, and {49}) (16/26 = 61.5%) with Intra.H. (Table 1). These progenies had statistically higher α values, ranging between 0.67051 and 0.68592 (0.6% to 2.8%). In contrast, three other progenies ({12}, {10}, and {8}) (3/26 = 11.5%) had statistically lower α values, ranging between 0.65302 and 0.66135 (2.0% to 0.8%) (Table 1). For progenies with Inter.H., evidence supporting H4 was observed in 15 progenies ({21}, {22}, {24}, {25}, {20}, {23}, {33}, {43}, {46}, {40}, {47}, {38}, {29}, {32}, and {37}) (15/29 = 44.8%). Their α values were statistically higher than the correction factor of 2/3, varying between 0.66996 and 0.68576 (0.4% to 2.8%) (Table 1). Conversely, five progenies ({35}, {36}, {44}, {45}, and {28}) (5/29 = 17.2%) had statistically lower α values, ranging between 0.65397 and 0.66081 (1.9% to 0.9%) (Table 1). Statistical evidence against H4 (meaning no statistical difference from α = 2/3) was observed in seven Intra.H. progenies (7/26= 26.9%), whereas this occurred in nine Inter.H. progenies (9/29= 31.0%) (Table 1). Contrary to the above findings for Pd, the 2/3 correction factor may provide a better estimate of L_S_ for more individual Inter.H. progenies than Intra.H. progenies. However, of all the progenies evaluated, 29.1% (16/55) were statistically equivalent to 2/3, which is 7.3% lower than that observed for Pd. In the consolidated results for hybridization types (Intra.H., Inter.H.) and Pd, the α parameter was statistically greater than the 2/3 correction factor, thus providing statistical evidence in favor of H4 (Table 1).

In Model 2 (M2: L_S_ = α Lβ + ɛ), the β parameter showed statistical evidence supporting fifth hypothesis (H5) in 10 progenies ({1}, {19}, {10}, {9}, {13}, {53}, {54}, {55}, {50}, and {51}) (10/26 = 38.5%) from the Intra.H. crossing. Their β values ranged from 2.06643 to 2.16317 (3.2% to 7.5% higher), which were statistically greater than 2 (“principle of similarity”). In contrast, for the remaining 16 progenies (16/26 = 61.5%), evidence against H5 was observed, as they showed no statistical difference from 2 (Table 2). In 12 ({27}, {20}, {23}, {33}, {39}, {31}, {35}, {48}, {47}, {34}, {37}, and {28}) (12/29 = 41.4%) of the 29 progenies with Inter.H. crosses, the β parameter showed statistical evidence supporting H5, which was indicated by values statistically greater than 2, ranging between 2.07134 and 2.20481 (3.4% and 9.3% higher) (Table 2). Conversely, for the remaining 15 progenies (17/29 = 58.6%), the absence of significant differences from 2 provided statistical evidence against H5 (Table 2). The “principle of similarity” (β = 2) was upheld in 60% (33/55) of the evaluated progenies, with a higher proportion among Intra.H. progenies than Inter.H. progenies. Consistent with this, the β parameter for the consolidated Intra.H. type showed statistical evidence against H5, indicating no significant difference from 2 (Table 2). In contrast, the β parameter for the consolidated Inter.H. type and the Pd showed evidence supporting H5, with statistically greater values than 2 (Table 2).

The root mean squared error (RMSE) for M1 were lower in the Intra.H. progenies (ranging from 1.126 to 1.955) compared with the Inter.H. progenies (ranging from 1.195 to 2.064), which is corroborated by examining the RMSE of the consolidated Intra.H. type (1.718) relative to the consolidated Inter.H. type (1.860) (Table 1). M2 showed a similar situation for the RMSE of the consolidated Intra.H. data (5.090) compared to the consolidated Inter.H. data (5.573) (Table 2) but with values 3.0 times greater than those found in M1 (Table 1). The RMSEs for M2 in the Intra.H. progenies (ranging from 2.948 to 7.083) fluctuated more than those in the Inter.H. progenies (ranging from 3.462 to 5.953) (Table 2). The RMSE of the Pd calculated for M1 (1.795) was three times lower than that obtained for M2 (5.383) (Table 1 and Table 2). M1 was more accurate than M2 for estimating L_S_.

### 2.2. Leaf Shape Characteristics Related to Leaf Size

Figure 1A illustrates the clustering of progenies from Intra.H. and Inter.H. crossing using the partitioning around medoids (PAM) method, which identified two distinct clusters. Cluster 1 comprised 38 progenies ({6}, {1}, {5}, {19}, {14}, {18}, {3}, {16}, {7}, {12}, {10}, {9}, {8}, {11}, {13}, {54}, {55}, {51}, {52}, {49}, {30}, {22}, {24}, {26}, {27}, {46}, {39}, {31}, {35}, {36}, {48}, {41}, {44}, {34}, {32}, {45}, {37}, and {28}), of which 20 progenies belonged to Intra.H. crosses (20/38= 52.6%) and 18 to Inter.H. crossing (18/38= 47.4%) (Figure 1A,F). Cluster 2 consisted of 17 progenies ({2}, {4}, {17}, {15}, {53}, {50}, {21}, {25}, {20}, {23}, {33}, {43}, {40}, {42}, {47}, {38}, and {29}), with six from Intra.H. crossing (6/17= 35.3%) and 11 from Inter.H. crossing (11/17= 64.7%) (Figure 1A,F). To visualize these clusters in a two-dimensional space, a principal component analysis was performed on the L_S_, the ratio of W to L (R_WL_), and ellipticity index (E_I_) variables, which successfully preserved 78.5% of the original data variability (Figure 1A). Compared with Cluster 1, Cluster 2 was characterized by higher values for L_S_ (8.5%), R_WL_ (5.3%), and E_I_ (1.4 percentage points) (Figure 1B–D), with statistically significant multivariate differences (Figure 1E). Figure 2 illustrates a visual example of the leaves classified into cluster 1 and cluster 2, along with their characterization by L_S_, R_WL_, and E_I_ (note that this does not account for leaf asymmetry, where one side of the leaf is larger than the other). Conversely, Intra.H. progenies showed no statistical differences compared with the Inter.H. progenies (T^2^ = 7.60581; *p* = 0.075; *n* = 55). The χ^2^ homogeneity of distribution test indicated that hybridization type had no influence on cluster formation; that is, the distribution was homogeneous (Figure 1F). Analysis of the results revealed that compared with the progenies in Cluster 1, the progenies in Cluster 2 presented larger leaves, a greater width-to-length ratio, and a surface area closer to that of an ideal ellipse. Cluster 2 was predominantly composed of Inter.H. progenies, whereas a more balanced proportion of both Intra.H. and Inter.H. progenies was found in Cluster 1. The multiple correlations between L_S_ and R_WL_ (γ = 0.11; *p* = 0.999; *n* = 55) and between L_S_ and E_I_ (γ = 0.19; *p* = 0.481; *n* = 55) were not statistically significant. However, the R_WL_ vs. E_I_ correlation (γ = 0.50; *p* < 0.001; *n* = 55) was positive but weakly statistically significant.

## 3. Discussion

Previous research on *C. arabica* has reported that the α parameter for M1 ranges between 0.64340 and 0.70125, with a mean of 0.66710 [15,38,44,45,46]. This range encompasses the α parameter values determined by M1 in this study for both Intra.H. and Inter.H. progenies (between 0.65302 and 0.68592), as well as those consolidated by hybridization type and Pd (between 0.67066 and 0.67118) (Table 1). These findings suggest that an α parameter value of 0.67000 in M1 could be suitable for estimating L_S_ at the species level (*C. arabica*) or for groups of progenies (Intra.H. and Inter.H.). This is evidenced by the lack of statistical differences in 36.4% of the cases when the α parameter in each evaluated progeny was compared to the α of Pd (α = 0.67110), as well as the lack of differences between Intra.H. and Inter.H. relative to the α of Pd (Table 1). Furthermore, a simulation assuming an α parameter of 0.67000 yielded an RMSE with reasonable accuracy (Appendix A) compared with the RMSE found per progeny (Table 1). This is further validated by the lack of statistical differences in 29.1% of the progenies compared with the correction factor of 2/3 (α = 0.66667) (Table 1). In *C. arabica*, an α parameter value of 0.667, estimated by Barros et al. [38], has shown strong performance in L_S_ estimation and consistency over time [10,42].

Reports exist of asymmetrically elliptical leaves, such as obovate or ovate leaves, where the base or apex is narrower than the opposite end, respectively [16]. In species with obovate leaves, the α parameter is 0.66, whereas for ovate leaves, it is 0.68; however, its value converges to 0.67 when considering similarities in the L:W ratio [18]. Furthermore, the statistical evidence regarding Pd in the progenies indicates that even a slight change in the α parameter values can affect the accurate estimation of L_S_ in specific progenies (Table 1). In this regard, the estimation of the α parameter for a particular progeny should ideally be obtained at the genotype level because of the subtle changes in S_h_ recorded by the R_WL_ and E_I_ variables in this study (Figure 1C,D). The α parameter could also vary because of the medial asymmetry of leaves, which is the difference between one side of the leaf relative to the central vein [36].

These findings have practical implications for the estimation of L_S_ in coffee mono-progeny and composite coffee varieties. For mono-progeny varieties such as Caturra, estimating a specific α parameter is crucial because of the potential variations in S_h_ observed in this study, leading to more accurate L_S_ estimations (Table 1, Figure 1C,D). Conversely, a general α parameter estimate of 0.67000 would be more suitable for composite varieties common in Colombia, such as Cenicafé 1 and Castillo^®^. This general value effectively captures the variability within mixed progenies, allowing accurate L_S_ estimation regardless of the specific S_h_, which cannot be precisely determined because of the randomness of genotypes in the field and during leaf sampling. The α parameter estimations found in this study for M1, as well as in previous coffee research [15,38,44,45,46], fall within the theoretical minimum range for triangular leaves (1/2) and the maximum for elliptical leaves (π/4) possible for M1 [54].

The absence of statistically significant differences (46.1%) between the Intra.H. progenies and Pd for M1, as opposed to 27.6% for Inter.H. progenies and Pd, might be linked to the lower genetic variability in Intra.H. progenies compared to Inter.H. progenies, which originate from crosses between *C. arabica* and *C. canephora* (Table 1). In progenies derived from Inter.H., after five or seven generations of self-pollination, the recombination of chromosomes from different genomes (*C. arabica* × *C. canephora*) results in significant variations in their phenotypic characteristics because of the presence of introgressed fragments. In contrast, Intra.H. progenies exhibit a narrow genetic base [55]. However, type Intra.H. showed no differences compared with the type Inter.H. in the Hotelling’s T^2^ test and confidence intervals (Table 1), which may be attributed to the extensive Arabization observed among the progenies in the filial generation (see in Section 4).

M2 was used to demonstrate the “principle of similarity” in coffee. The results revealed that in 60% of the evaluated progenies, L_S_ was proportional to the square of L (Table 2). However, analysis of the β parameter in the groupings indicated that only the Intra.H. type was statistically equal to 2, unlike the Inter.H. type and the Pd, which were not (Table 2). These observations suggest that the “principle of similarity” is applicable at the progeny level but fails to be consistent at the species or composite variety level, as was proposed with the α parameter of M1. In *Quercus pannosa*, variation in S_h_ (E_I_ different from 1) did not affect the validity of the Montgomery model (M1) for calculating L_S_, although it did impact the “principle of similarity”, which showed values below 2 [24]. This implies that M2 is most useful for estimating L_S_ in mono-progeny coffee varieties. However, for sets of progenies (Intra.H. or Inter.H.) With a β parameter significantly different from 2, depending on the composition, M2 would not be the most appropriate choice. An example is the M2 developed for the composite variety Castillo^®^, which exhibited a β parameter statistically greater than 2 (95% CI: 2.052–2.080; *n* = 6441) [14]. On the other hand, a group with a greater number of progenies whose β parameter is statistically equal to 2 may indicate less variability in S_h_, as observed in Intra.H. progenies (69.2%) compared to Inter.H. progenies (51.7%) (Table 2). L_S_ is proportional to the square of L in bamboo species when there is low intraspecific variation in S_h_ [56]. M2 allows for the more straightforward calculation of L_S_ in the field because it only requires one leaf dimension, but it is less accurate than M1 is (Table 2). In studies involving a large number of leaves, single-dimensional leaf models can simplify the measurement process at the cost of greater error relative to the true value [14].

In this study, the RMSE values found for M1 (Table 1) are comparable to those obtained in a previous study involving eight *C. arabica* genotypes in Ethiopia [44], where M1 showed similar RMSE values (between 1.126 and 2.064). In *C. arabica*, M1 has consistently demonstrated its accuracy in calculating L_S_ [15,38,44,45,46]. This makes it a valuable tool for field studies that require non-destructive methods [16,54]. M1 has also performed well in other species, including grapevines [9], bamboo [16], Michelia [57], and alpine oak [24]. Across various studies, M1 has consistently proven more accurate than M2 for estimating L_S_ [9,15].

Based on the foregoing, the main points to consider when selecting between the Montgomery model and the principle of similarity are summarized in Table 3. The application of these models in coffee makes it possible to study the source-sink relationship in different genotypes, estimate the light extinction coefficient in canopies with varied architectures, and evaluate the impact of defoliation from mechanical or biological causes [50,58,59].

The PAM analysis of S_h_ attributes (R_WL_ and E_I_) and L_S_ revealed a distinct grouping of progenies within the clusters, which are statistically different (Figure 1A,E). Cluster 1, comprising 69.1% of the total progenies evaluated, exhibited a relatively balanced composition, with 52.6% Intra.H. progenies and 47.4% Inter.H. progenies (Figure 1F). In contrast, Cluster 2, encompassing the remaining 30.9% of the progenies, showed a predominance of Inter.H. progenies (64.7%) over Intra.H. progenies (35.3%) (Figure 1F) and was characterized by the highest L_S_, R_WL_, and E_I_ values compared with those of Cluster 1 (Figure 1B,D). This suggests that Cluster 2 is more closely linked to the characteristics of Inter.H. progenies and, very likely, to some attributes of *C. canephora*, a species known for having a relatively large L_S_ and an E_I_ closer to 1. Although they are not exclusive of Intra.H., as the cluster composition displays and the χ^2^ homogeneity of distribution (Figure 1A,E,F). This indicates that progenies from both intraspecific (Intra.H.) and interspecific (Inter.H.) hybrids can exhibit common Sh characteristics, which in the present case are differentiated by the cluster. For instance, the L_S_ in 10 *C. arabica* genotypes (28.45 and 60.85 cm^2^) was smaller than that in 10 *C. canephora* genotypes (23.77 and 71.21 cm^2^) [42]. Antunes et al. [15] generally reported that *C. canephora* leaves are larger than *C. arabica* leaves. M1 has an α parameter value of 0.7100 in *C. canephora*, which translates to an E_I_ of 0.90 (closer to 1) [60]. In contrast, the estimated average α parameter for *C. arabica* from previous research is 0.6671, corresponding to an E_I_ of 0.85 [15,38,44,45,46].

The positive Spearman correlation between R_WL_ and E_I_ in this study, along with the higher R_WL_ values (4.2%) and E_I_ values (1 percentage point) in Cluster 2 than in Cluster 1 (Figure 1C,D), align with the literature (leaves with a greater R_WL_ are also more elliptical). In grapevines, higher R_WL_ values have been associated with larger α parameters in M1, leading to higher E_I_ values [9]. However, the present study did not report a negative correlation between R_WL_ and L_S_ for 12 tree species [16].

## 4. Materials and Methods

### 4.1. Location and Plant Material

This study was conducted at the Naranjal Experimental Station (Chinchiná, Caldas, Colombia; 4°58′19.1″ N, 75°39′8.2″ W, 1407 m), which reflects the typical climatic and soil conditions of Colombia’s central coffee-growing region [61]. In November 2020, 55 advanced progenies from the coffee breeding program were established in experimental plots consisting of seven plants (five effective plants and two border plants) under a block design with five levels of blocking associated with the slope. A planting density of 7142 plants/ha (1.4 m between rows and 1.0 m between plants, arranged in a rectangle) was used. The progenies evaluated correspond to intraspecific *C. arabica* hybridization crosses (Intra.H.) and an interspecific hybridization cross group between *C. arabica* and *C. canephora* (Inter.H.) with a high degree of Arabization (Table 4). Throughout the phenological cycle, soil fertilization was performed according to the nutritional requirements of the crop [62], while the integrated management of weeds, pests, and diseases followed the technical recommendations established by Cenicafé [63]. Between September and October 2023, 46 leaves of different sizes were randomly collected from each experimental plot, totaling 228 leaves per progeny. The samples were subsequently stored in zip-lock bags at 5 °C to maintain turgidity until processing (no longer than 4 days).

### 4.2. Leaf Image Processing

In the laboratory, the petiole of each leaf was carefully dissected before scanning. The abaxial side of the leaves was then digitized into images using a CanoScan LiDE 300 scanner (Canon, Tokyo, Japan) at a resolution of 300 dpi. From these images, the Cartesian coordinates of each leaf were extracted using the “pliman” package [13]. Using these coordinates, the leaf length (L; cm), defined as the distance from the petiole attachment point to the apex [15]; the leaf width (W; cm), defined as the maximum distance between two points on the leaf perimeter forming a perpendicular with the progeny connecting the leaf base and apex [64]; and the leaf size (L_S_; cm^2^) were calculated with the “biogeom” package [65]. L, W, and L_S_ were recorded for 12,540 leaves. Additionally, the ratio of W to L (R_WL_) was calculated.

### 4.3. Parameter Estimation and Hypothesis Testing

Allometric indicators associated with the LS variable, referencing changes in L and/or W, were calculated using two models. Model 1 (M1: L_S_ = α LW + ɛ), also known as the “Montgomery model,” assumes that LS is proportional to a rectangle with sides L and W multiplied by a normalization constant α [16]. Model 2 (M2: L_S_ = α L^β^ + ɛ) assumes a power law relationship between L_S_ and L, with parameters α and β [9]. When the parameter β equals 2, L_S_ is proportional to the square of L and adheres to the “principle of similarity” [19,24]. Confidence intervals for α and β in both models were constructed for each progeny, hybridization type, and grouped data (Pd) using the bootstrap percentile method. This involved nonlinear regression algorithms (B = 10,000) with the “minpack.lm” [66] and “car” [67] packages.

In M1, four research hypotheses were evaluated for the parameter α: (a) the first hypothesis was evaluated by comparing the confidence intervals for each progeny and hybridization type against the confidence interval of the grouped data (Pd) [$H_{1}:\mu_{\alpha_{i}}-\mu_{\alpha_{Pd}}\neq 0$, where i is the i-th progeny or hybridization type]; (b) for the second hypothesis, we assessed whether the confidence intervals for each progeny, hybridization type, and Pd did not contain the lowest α reported for *C. arabica* [44] [$H_{2}:\;\mu_{\alpha_{i}}\;\neq 0.64340$; where i is the i-th progeny, hybridization type, or Pd]; (c) for the third hypothesis, we examined whether the confidence intervals for each progeny, hybridization type, and Pd did not contain the highest α reported for combined *Coffea* data with intraspecific and interspecific hybridization [15] [$H_{3}:\;\mu_{\alpha_{i}}\;\neq 0.70125$; where i is the i-th progeny, hybridization type, or Pd]; and (d) for the fourth hypothesis, we determined whether the confidence intervals for each progeny, hybridization type, and Pd did not contain the correction factor of 2/3 [18,22] [$H_{4}:\;\mu_{\alpha_{i}}\;\neq 2/3$; where i is the i-th progeny, hybridization type, or Pd]. Similarly, for Model 2 (M2), only the fifth hypothesis was evaluated. This hypothesis determined if the confidence intervals for each progeny, hybridization type, and Pd did not contain a value of 2 [$H_{5}:\;\mu_{\alpha_{i}}\;\neq 2$, where i is the i-th progeny, hybridization type, or Pd], to verify the “principle of similarity” [19]. Decisions were made on the basis of the probability values concerning the confidence intervals, following the methodology proposed by Rousselet et al. [68]. Independence was assumed between the progeny samplings; hence, no adjustments were made to the number of tests involved. Additionally, the root mean squared error (RMSE) was calculated to measure the accuracy between the observed L_S_ values and the estimated leaf size (L^_S_) values from each model, following the methodology of Walther and Moore [69] and using the “Metrics” package [70].

### 4.4. Analysis of LS and Sh

The average values of L_S_ and R_WL_ per progeny and their corresponding α parameters from M1 were used for the inferential component. The α parameter was expressed in terms of the ellipticity index (E_I_), defined as E_I_ = 4 α/π, to quantify the deviation of L_S_ about the area of an ideal ellipse [21]. Multivariate statistical differences for L_S_, R_WL_, and E_I_ between progenies from Intra.H. and Inter.H. crosses were evaluated using Hotelling’s T^2^ test after validating the assumptions of multivariate normality (Mardia’s test) and homogeneity in the covariance matrix (Box’s M test). Then, possible groupings independent of the Intra.H. and Inter.H. crossing were evaluated using the partitioning around medoids (PAM) clustering method, and a principal component analysis was implemented on the variables to represent them in a two-dimensional plane. The number of clusters was defined using the “majority rule” by evaluating 30 methods (silhouette, gap, d-index, etc.) [71] and by examining the cluster stability index with bootstrapping [72]. A second Hotelling’s T^2^ test was performed to compare the mean vectors of the L_S_, R_WL_, and E_I_ variables for the two resulting groups. To determine whether the cluster classification changed the original proportions of progenies by crossing type (Intra.H. and Inter.H.), a χ^2^ homogeneity of distribution test was conducted. Spearman’s multiple correlations were adjusted using the Bonferroni method for the L_S_, R_WL_, and E_I_ variables. This section utilized the “MVTest” [73], “MVN” [74], “biotools” [75], “cluster” [76], “factoextra” [77], “NbClust” [71], “fpc” [72], “ggstatsplot” [78], “correlation” [79], “ggplot2” [80] and “scatterplot3d” [81] packages. All the packages were executed in R software version 4.4.1 [82].

## 5. Conclusions

Our findings indicate that L and W adequately estimate L_S_ for all 55 coffee progenies evaluated, offering a simple and non-destructive method that streamlines field data collection and enables consecutive measurements over time. M1, which is based on L and W, demonstrated good accuracy in calculating L_S_ and proved valid for both Intra.H. and Intra.H. progenies. This suggests its suitability for estimating L_S_ in mono-progeny varieties. Furthermore, M1, with a general α parameter of 0.67, could be appropriate for groups of progenies, such as composite varieties, and could even be representational at the species level while maintaining satisfactory accuracy. None of the progenies evaluated with M1 presented α values less than 0.64340 or greater than 0.70125, the extreme values reported in the literature for *C. arabica*. Conversely, the “principle of similarity”, evaluated using M2, showed that the proportionality between L_S_ and the square of L holds only at the progeny level for coffee. Consequently, the “principle of similarity” cannot be generalized to the species or composite variety level, unlike what was proposed with M1. Therefore, M2 can be used to calculate L_S_ in mono-progeny varieties with an accuracy lower than that achieved with M1. The S_h_ characteristics (R_WL_ and E_I_) and L_S_ can vary between Intra.H. and Inter.H. progenies, influencing the parameter fitting in both M1 and M2. In coffee, leaves with a higher R_WL_ were more elliptical. The impact of S_h_ variation on the accuracy of L_S_ estimation models warrants further investigation, particularly focusing on differences in leaf surface stemming from the central vein.

## Figures and Tables

**Figure 1 plants-14-02985-f001:**
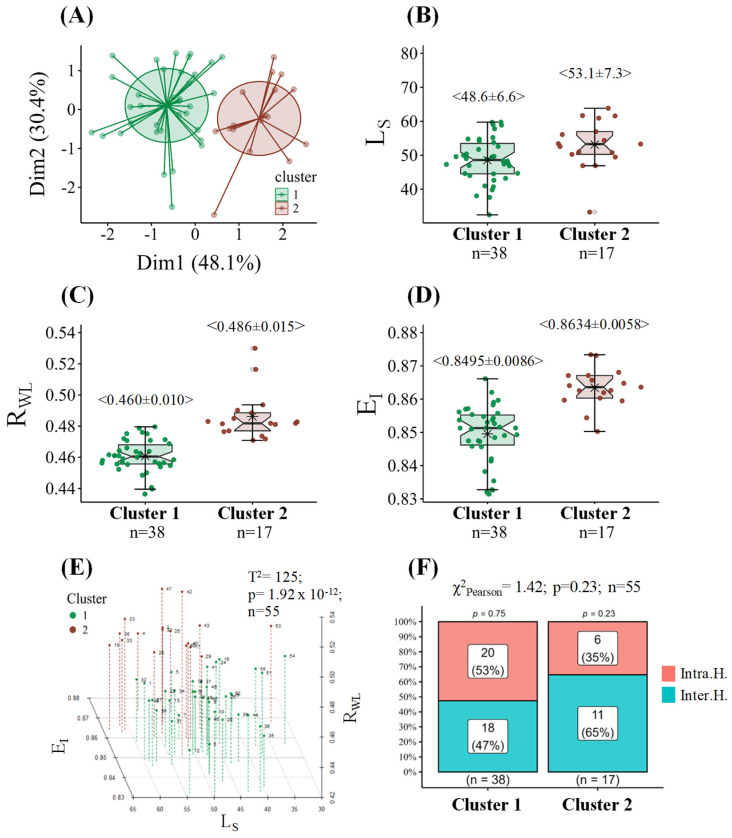
Two-dimensional representation of the 36 progenies forming Cluster 1 and the 19 progenies forming Cluster 2 [Dimensions 1 and 2 collectively explain 77.7% of the total variance] (**A**). The leaf size (L_S_: cm^2^) (**B**), width-to-length ratio (R_WL_) (**C**), and ellipticity index (E_I_) (**D**) distributions for the progenies within both clusters are shown. Representation of a 3D plot illustrating the L_S_, R_WL_, and E_I_ variables for the clusters, with the results of Hotelling’s T2 multivariate test (**E**). The proportion of progenies with intraspecific *C. arabica* hybridization (Intra.H.) and interspecific *C. arabica* with *C. canephora* hybridization (Inter.H.) per cluster, with the corresponding χ^2^ test for homogeneity of distribution (**F**). On the box plots, values within < > represent the mean ± standard deviation (SD), and the asterisk ✳ indicates their position. The numbers on the 3D graph correspond to the numerical treatment identifying each progeny (see in Section 4).

**Figure 2 plants-14-02985-f002:**
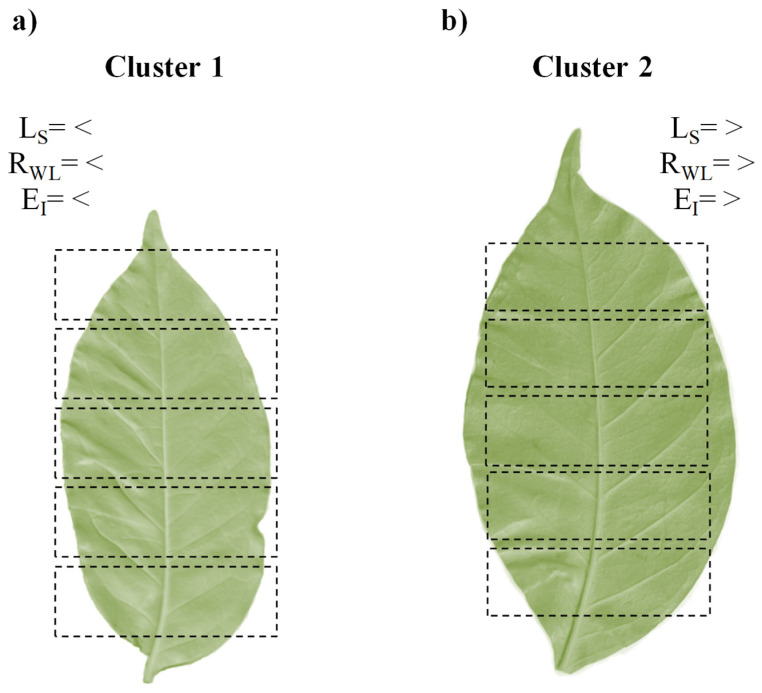
Graphical example of leaves classified into cluster 1 (**a**) and cluster 2 (**b**), with the respective conceptual differences for the characteristics leaf size (L_S_: cm^2^), width-to-length ratio (R_WL_), and ellipticity index (E_I_). The rectangles serve as a reference to observe the changes in size and shape in each leaf; each rectangle is 5.1 cm in width and 1.6 cm in height.

**Table 1 plants-14-02985-t001:** The α parameter values estimated using Model 1 (M1), with 95% confidence intervals (LCI–UCI). The *p* values for hypotheses $H_{1}:\mu_{\alpha_{i}}-\mu_{\alpha_{Pd}}\neq 0$, $H_{2}:\;\mu_{\alpha_{i}}\;\neq 0.64340$, $H_{3}:\;\mu_{\alpha_{i}}\;\neq 0.70125$, and $H_{4}:\;\mu_{\alpha_{i}}\;q\neq 2/3$ and the root mean square error (RMSE) for M1 are also provided. In the first column, labels denote the progeny ({progeny treatment number}), hybridization type (*H*), and grouped data (Pd) per model. Intra.H. represents the consolidated data for all intraspecific *C. arabica* hybridization progenies, while Inter.H. represents the consolidated data for all interspecific *C. arabica* with *C. canephora* hybridization progenies. In the model, the response variable is the leaf size (L_S_: cm^2^), and the explanatory variable is the rectangular area (LW: cm^2^), which is equivalent to the leaf length multiplied by the width from the training data. The symbol ∈ indicates that the value falls within the confidence interval, whereas ∉ indicates that it does not. Values above the confidence interval are highlighted in green, and those below are highlighted in red.

{#} | *H* | Pd	M1: L_S_ = α LW + ɛ					
	α (LCI–UCI)	H_1_	H_2_	H_3_	H_4_	RMSE
{1}	0.66686 (0.66368–0.67003)	∈	∈	∈	∉	1.955
{2}	0.68176 (0.67830–0.68516)	∈	∈	∈	∈	1.839
{3}	0.67095 (0.66769–0.67421)	∉	∈	∈	∈	1.576
{4}	0.68592 (0.68267–0.68910)	∈	∈	∈	∈	1.834
{5}	0.67704 (0.67380–0.68028)	∈	∈	∈	∈	1.781
{6}	0.68024 (0.67701–0.68353)	∈	∈	∈	∈	1.329
{7}	0.67469 (0.67150–0.67782)	∈	∈	∈	∈	1.745
{8}	0.66135 (0.65833–0.66430)	∈	∈	∈	∈	1.573
{9}	0.66838 (0.66541–0.67126)	∉	∈	∈	∉	1.554
{10}	0.65302 (0.65010–0.65602)	∈	∈	∈	∈	1.590
{11}	0.67051 (0.66716–0.67392)	∉	∈	∈	∈	1.838
{12}	0.65352 (0.65024–0.65669)	∈	∈	∈	∈	1.799
{13}	0.66858 (0.66576–0.67137)	∉	∈	∈	∉	1.652
{14}	0.67051 (0.66766–0.67329)	∉	∈	∈	∈	1.443
{15}	0.67828 (0.67573–0.68081)	∈	∈	∈	∈	1.736
{16}	0.67320 (0.67022–0.67618)	∉	∈	∈	∈	1.380
{17}	0.67504 (0.67140–0.67858)	∈	∈	∈	∈	1.858
{18}	0.66882 (0.66631–0.67135)	∉	∈	∈	∉	1.326
{19}	0.66967 (0.66646–0.67303)	∉	∈	∈	∉	1.567
{49}	0.67306 (0.66935–0.67665)	∉	∈	∈	∈	1.542
{50}	0.67520 (0.67175–0.67865)	∈	∈	∈	∈	1.664
{51}	0.66842 (0.66542–0.67139)	∉	∈	∈	∉	1.207
{52}	0.66561 (0.66214–0.66900)	∈	∈	∈	∉	1.489
{53}	0.68104 (0.67774–0.68429)	∈	∈	∈	∈	1.126
{54}	0.67241 (0.66861–0.67621)	∉	∈	∈	∈	1.300
{55}	0.67180 (0.66802–0.67559)	∉	∈	∈	∈	1.438
Intra.H. [*n* = 5928]	0.67118 (0.67049–0.67187)	∉	∈	∈	∈	1.718
{20}	0.67568 (0.67261–0.67865)	∈	∈	∈	∈	1.828
{21}	0.68101 (0.67761–0.68445)	∈	∈	∈	∈	1.845
{22}	0.67209 (0.66874–0.67548)	∉	∈	∈	∈	1.855
{23}	0.67863 (0.67525–0.68202)	∈	∈	∈	∈	2.064
{24}	0.67517 (0.67130–0.67910)	∈	∈	∈	∈	1.711
{25}	0.68576 (0.68222–0.68931)	∈	∈	∈	∈	1.819
{26}	0.66703 (0.66277–0.67126)	∉	∈	∈	∉	1.948
{27}	0.66431 (0.66119–0.66739)	∈	∈	∈	∉	1.989
{28}	0.65832 (0.65432–0.66221)	∈	∈	∈	∈	1.809
{29}	0.67744 (0.67367–0.68121)	∈	∈	∈	∈	1.748
{30}	0.66558 (0.66251–0.66869)	∈	∈	∈	∉	1.801
{31}	0.66418 (0.66126–0.66703)	∈	∈	∈	∉	1.708
{32}	0.67406 (0.67072–0.67732)	∉	∈	∈	∈	1.848
{33}	0.67750 (0.67442–0.68058)	∈	∈	∈	∈	1.999
{34}	0.66868 (0.66539–0.67199)	∉	∈	∈	∉	1.856
{35}	0.65613 (0.65126–0.66010)	∈	∈	∈	∈	1.772
{36}	0.65404 (0.65088–0.65720)	∈	∈	∈	∈	1.376
{37}	0.67133 (0.66724–0.67545)	∉	∈	∈	∈	1.912
{38}	0.67992 (0.67723–0.68262)	∈	∈	∈	∈	1.767
{39}	0.66546 (0.66108–0.67012)	∈	∈	∈	∉	1.764
{40}	0.67104 (0.66791–0.67413)	∉	∈	∈	∈	1.550
{41}	0.66863 (0.66574–0.67140)	∉	∈	∈	∉	1.456
{42}	0.66780 (0.66504–0.67057)	∈	∈	∈	∉	1.568
{43}	0.67918 (0.67662–0.68180)	∈	∈	∈	∈	1.195
{44}	0.65397 (0.65086–0.65700)	∈	∈	∈	∈	1.398
{45}	0.66081 (0.65760–0.66402)	∈	∈	∈	∈	1.629
{46}	0.66996 (0.66704–0.67289)	∉	∈	∈	∈	1.428
{47}	0.67696 (0.67418–0.67983)	∈	∈	∈	∈	1.652
{48}	0.66636 (0.66347–0.66915)	∈	∈	∈	∉	1.802
Inter.H. [*n* = 6612]	0.67066 (0.66998–0.67130)	∉	∈	∈	∈	1.860
Pd [*n* = 12,540]	0.67089 (0.67041–0.67136)	∉	∈	∈	∈	1.795

**Table 2 plants-14-02985-t002:** The α and β parameter values estimated using Model 2 (M2), with 95% confidence intervals (LCI–UCI). The *p* values for the hypothesis $H_{5}:\;\mu_{\alpha_{i}}\;\neq 2$ and the root mean square error (RMSE) for M2 are also provided. In the first column, labels denote the progeny ({progeny treatment number}), hybridization type (*H*), and grouped data (Pd) per model. Intra.H. represents the consolidated data for all intraspecific *C. arabica* hybridization progenies, while Inter.H. represents the consolidated data for all interspecific *C. arabica* with *C. canephora* hybridization progenies. In the model, the response variable is leaf size (L_S_: cm^2^), and the explanatory variable is the rectangular area (LW: cm^2^), which is equivalent to the leaf length multiplied by the width from the training data. The symbol ∈ indicates that the value falls within the confidence interval, whereas ∉ indicates that it does not. Values above the confidence interval are highlighted in green.

{#} | *H* | Pd	M2: L_S_ = α L^β^ + ɛ			
	α (LCI–UCI)	β (LCI–UCI)	H_5_	RMSE
{1}	0.25877 (0.22601–0.29345)	2.07500 (2.02716–2.12488)	∈	4.199
{2}	0.36990 (0.29479–0.45083)	1.96047 (1.88175–2.04577)	∉	5.715
{3}	0.31694 (0.26548–0.37583)	1.99277 (1.92571–2.05904)	∉	4.804
{4}	0.38739 (0.28614–0.50618)	1.93773 (1.83103–2.05025)	∉	7.083
{5}	0.35360 (0.28476–0.43064)	1.95203 (1.87291–2.03416)	∉	5.989
{6}	0.34285 (0.26477–0.42618)	1.94735 (1.85676–2.04846)	∉	4.573
{7}	0.27203 (0.22436–0.32807)	2.03317 (1.95950–2.10516)	∉	5.156
{8}	0.31192 (0.24560–0.38438)	1.97482 (1.89293–2.06343)	∉	5.511
{9}	0.22642 (0.19681–0.26117)	2.11907 (2.06329–2.17123)	∈	3.882
{10}	0.24628 (0.20807–0.28729)	2.09340 (2.03287–2.15745)	∈	4.245
{11}	0.31048 (0.25957–0.36374)	2.00134 (1.93963–2.06942)	∉	5.064
{12}	0.30810 (0.25019–0.37304)	1.98809 (1.91314–2.06586)	∉	4.751
{13}	0.25891 (0.21686–0.30564)	2.06643 (2.00088–2.13351)	∈	4.723
{14}	0.27174 (0.22945–0.31864)	2.05013 (1.98827–2.11394)	∉	4.409
{15}	0.28537 (0.24248–0.33555)	2.05120 (1.98998–2.11073)	∉	5.305
{16}	0.28925 (0.24669–0.33575)	2.04323 (1.98315–2.10506)	∉	3.989
{17}	0.28314 (0.23382–0.33765)	2.05854 (1.98845–2.13224)	∉	4.963
{18}	0.28766 (0.24146–0.33643)	2.03916 (1.97773–2.10566)	∉	4.324
{19}	0.20524 (0.17186–0.24022)	2.16050 (2.09827–2.22836)	∈	4.015
{49}	0.28495 (0.21925–0.35683)	2.03631 (1.94577–2.13661)	∉	4.932
{50}	0.24479 (0.21155–0.28038)	2.11315 (2.05973–2.16877)	∈	3.994
{51}	0.22530 (0.19806–0.25448)	2.14276 (2.09190–2.19486)	∈	2.948
{52}	0.28975 (0.24273–0.34120)	2.02721 (1.96067–2.09671)	∉	4.171
{53}	0.22691 (0.18636–0.27136)	2.16317 (2.08460–2.24513)	∈	3.214
{54}	0.24731 (0.20467–0.29337)	2.11482 (2.04146–2.19285)	∈	3.152
{55}	0.21637 (0.17793–0.25872)	2.16075 (2.08509–2.23968)	∈	3.468
Intra.H. [*n* = 5928]	0.30201 (0.29032–0.31374)	2.01444 (1.99948–2.02970)	∉	5.090
{20}	0.24093 (0.19998–0.28392)	2.11504 (2.05074–2.18478)	∈	5.139
{21}	0.27413 (0.22135–0.33404)	2.07572 (1.99774–2.15567)	∉	5.166
{22}	0.30906 (0.25770–0.36612)	2.00135 (1.93558–2.06985)	∉	5.467
{23}	0.28115 (0.23273–0.33223)	2.07134 (2.00753–2.14122)	∈	5.353
{24}	0.29412 (0.25208–0.33994)	2.03489 (1.97629–2.09582)	∉	4.258
{25}	0.34505 (0.27538–0.42378)	1.98475 (1.90178–2.07040)	∉	5.396
{26}	0.27678 (0.22209–0.33890)	2.04462 (1.96343–2.12882)	∉	5.651
{27}	0.25509 (0.20864–0.30927)	2.07474 (2.00025–2.14887)	∈	5.866
{28}	0.22551 (0.18432–0.27515)	2.12108 (2.04049–2.19718)	∈	4.900
{29}	0.28506 (0.23698–0.33491)	2.05095 (1.98608–2.12277)	∉	5.063
{30}	0.28951 (0.23539–0.35113)	2.02048 (1.94664–2.09670)	∉	5.953
{31}	0.22457 (0.19007–0.26139)	2.11258 (2.05475–2.17429)	∈	5.050
{32}	0.30869 (0.26549–0.35446)	2.00993 (1.95822–2.06454)	∉	5.213
{33}	0.22501 (0.19468–0.25797)	2.14477 (2.09394–2.19643)	∈	5.339
{34}	0.18521 (0.14555–0.22895)	2.20373 (2.12046–2.29300)	∈	5.111
{35}	0.21729 (0.17268–0.26875)	2.13535 (2.04595–2.22539)	∈	3.794
{36}	0.27029 (0.22599–0.31829)	2.05141 (1.98538–2.12233)	∉	3.793
{37}	0.18678 (0.15636–0.22212)	2.20481 (2.13550–2.27293)	∈	4.874
{38}	0.28646 (0.24083–0.33785)	2.05624 (1.99237–2.12073)	∉	5.345
{39}	0.21582 (0.17874–0.25908)	2.13592 (2.06139–2.20890)	∈	4.531
{40}	0.40067 (0.31984–0.48698)	1.92734 (1.84768–2.01528)	∉	4.984
{41}	0.30507 (0.26861–0.34154)	2.02238 (1.97772–2.07172)	∉	3.462
{42}	0.36245 (0.30029–0.43098)	1.99611 (1.92690–2.06835)	∉	4.917
{43}	0.35200 (0.30674–0.40401)	1.98069 (1.92476–2.03393)	∉	3.677
{44}	0.29143 (0.25033–0.33579)	2.02361 (1.96545–2.08349)	∉	3.642
{45}	0.26744 (0.22596–0.31237)	2.05124 (1.99168–2.11369)	∉	5.206
{46}	0.28228 (0.24817–0.31667)	2.03401 (1.99003–2.08263)	∉	3.939
{47}	0.26209 (0.21597–0.31384)	2.11646 (2.04455–2.19042)	∈	5.032
{48}	0.24343 (0.20059–0.29256)	2.08918 (2.01817–2.15936)	∈	5.378
Inter.H. [*n* = 6612]	0.28027 (0.26962–0.29070)	2.05001 (2.03580–2.06499)	∈	5.573
Pd [*n* = 12,540]	0.28729 (0.27967–0.29518)	2.03737 (2.02691–2.04776)	∈	5.383

**Table 3 plants-14-02985-t003:** Criteria for selecting the Montgomery model (M1) or the principle of similarity (M2) for estimating leaf size in the field of progenies/groups with intraspecific *C. arabica* hybridization (Intra.H.), interspecific *C. arabica* with *C. canephora* hybridization (Inter.H.), and pooled data (Pd). Leaf length (L) and leaf width (W). In Accuracy High, it refers to twice the RMSE values. In speed of measure, fast refers to one measure vs. two measures.

	Intra.H.		Inter.H.		Pd	
	M1	M2	M1	M2	M1	M2
Estimation of LS to progenies	Yes	Yes	Yes	Yes	-	-
Estimation of LS for the group	Yes	Yes	Yes	Not	Yes	Not
Accuracy	High	Low	High	Low	High	Low
Leaf dimensions in the measure	L and W	L	L and W	L	L and W	L
Speed of measure in the field	Slow	Fast	Slow	Fast	Slow	Fast

**Table 4 plants-14-02985-t004:** Progeny {treatment number identifying the progeny} and crosses from intraspecific *C. arabica* hybridization (Intra.H.) and interspecific *C. arabica* and *C. canephora* hybridization (Inter.H.). Advancement in the filial generation (F).

Hybridization | Progeny	{#}	Crossing	F
Intra.H.			
MEG105003(2017-1) #251	{6}	(Caturra × Timor hybrid) × Etiopía	F5
MEG102014(2017-2) #295	{1}	Etiopía × (Caturra × Timor hybrid)	F5
MEG102014(2017-2) #721	{2}	Etiopía × (Caturra × Timor hybrid)	F5
MEG102004(2010-6) #150	{5}	(Caturra × Timor hybrid) × Timor hybrid	F5
MEG102004(2010-6) #47	{4}	Timor hybrid × (Caturra × Timor hybrid)	F5
MEG105001(LIBANO 7 × 7) #1359	{19}	Caturra × Timor hybrid	F8
MEG105001(LIBANO 7 × 7) #1472	{14}	Caturra × Timor hybrid	F7
MEG105001(LIBANO 8 × 8) #195	{18}	Caturra × Timor hybrid	F8
MEG105001(LIBANO 8 × 8) #433	{17}	Caturra × Timor hybrid	F8
MEG102014(2017-2) #1949	{3}	Caturra × Timor hybrid	F7
MEG105001(2013-2) #48	{15}	Caturra × Timor hybrid	F8
MEG105001(2013-2) #706	{16}	Caturra × Timor hybrid	F8
MEG105001(LIBANO 8 × 8) #304	{7}	[Caturra × (Caturra × *C. Canephora*)] × [Catuaí × (Caturra × Borbón)]	F5
MEG105001(LIBANO 8 × 8) #326	{12}	[(Caturra × Timor hybrid) × (Caturra × Timor hybrid)] × [Catuaí × (Caturra × Borbón)]	F5
MEG105001(LIBANO 8 × 8) #349	{10}	[Caturra × (Caturra × *C. Canephora*)] × Etiopia	F5
MEG105001(LIBANO 8 × 8) #380	{9}	[Caturra × (Caturra × *C. Canephora*)] × [Catuaí × (Caturra × Borbón)]	F5
MEG105001(LIBANO 8 × 8) #407	{8}	[(Caturra × Timor hybrid) × (Caturra × Timor hybrid)] × (Sudán Rume × Catuaí)	F5
MEG105001(LIBANO 8 × 8) #568	{11}	(Caturra × Timor hybrid) × Dalecho	F5
MEG105001(LIBANO 8 × 8) #571	{13}	[(Caturra × Timor hybrid) × (Caturra × Timor hybrid)] × Etiopia	F5
CU1819	{53}	Caturra × Timor hybrid	F5
CU1825	{54}	Caturra × Timor hybrid	F5
CU1849	{55}	Caturra × Timor hybrid	F5
CU1953	{50}	Caturra × Timor hybrid	F5
CU2021	{51}	Caturra × Timor hybrid	F5
CU2034	{52}	Caturra × Timor hybrid	F5
CX2866	{49}	Caturra × Timor hybrid	F5
Inter.H.			
MEG105001 BLONAY #170,173	{30}	(Caturra × *C. canephora*) × Caturra	F6
MEG105001(LIBANO 8 × 8) #123	{21}	Caturra × [(Caturra × *C. canephora*) × Caturra]	F5
MEG105001(LIBANO 8 × 8) #139	{22}	(Caturra × *C. canephora*) × Caturra	F7
MEG105001(LIBANO 8 × 8) #159	{24}	(Caturra × *C. canephora*) × Caturra	F7
MEG105001(LIBANO 8 × 8) #290	{26}	(Caturra × *C. canephora*) × Caturra	F7
MEG105001(LIBANO 8 × 8) #365	{25}	(Caturra × *C. canephora*) × Caturra	F7
MEG105001(LIBANO 8 × 8) #469	{27}	(Caturra × *C. canephora*) × Caturra	F7
MEG105001(LIBANO 8 × 8) #601	{20}	Caturra × [(Caturra × *C. canephora*) × Caturra]	F5
MEG105001(LIBANO 8 × 8) #615	{23}	(Caturra × *C. canephora*) × Caturra	F7
MEG105001(2013-2) #102	{33}	(Caturra × *C. canephora*) × Caturra	F6
MEG102003(2009-17) #109	{43}	(Caturra × *C. canephora*) × Caturra	F5
MEG102003(2009-17) #13	{46}	(Caturra × *C. canephora*) × Caturra	F5
MEG105001(2013-3) #1511	{39}	(Caturra × *C. canephora*) × Caturra	F6
MEG105001(2013-2) #165	{31}	(Caturra × *C. canephora*) × Caturra	F6
MEG102003(2009-17) #250	{40}	(Caturra × *C. canephora*) × Caturra	F5
MEG105001(2013-2) #289	{35}	(Caturra × *C. canephora*) × Caturra	F6
MEG102003(2009-17) #300	{42}	(Caturra × *C. canephora*) × Caturra	F5
MEG105001(2013-2) #552	{36}	(Caturra × *C. canephora*) × Caturra	F6
MEG102003(2009-17) #561	{48}	(Caturra × *C. canephora*) × Caturra	F5
MEG102003(2009-17) #572	{41}	(Caturra × *C. canephora*) × Caturra	F5
MEG102003(2009-17) #583	{47}	(Caturra × *C. canephora*) × Caturra	F5
MEG102003(2009-17) #601	{44}	(Caturra × *C. canephora*) × Caturra	F5
MEG105001(2013-2) #679	{38}	(Caturra × *C. canephora*) × Caturra	F6
MEG105001(2013-2) #698	{29}	(Caturra × *C. canephora*) × Caturra	F6
MEG105001(2013-2) # 718	{34}	(Caturra × *C. canephora*) × Caturra	F6
MEG105001(2013-2) #84	{32}	(Caturra × *C. canephora*) × Caturra	F6
MEG102003(2009-17) #86	{45}	(Caturra × *C. canephora*) × Caturra	F5
MEG105001(2013-2) #98	{37}	(Caturra × *C. canephora*) × Caturra	F6
MEG105001(2013-2) #305,493	{28}	(Caturra × *C. canephora*) × Caturra	F6

## Data Availability

Data will be made available on reasonable request.

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
