# Peer review of "Non-Destructive Estimation of Leaf Size and Shape Characteristics in Advanced Progenies of *Coffea arabica* L. from Intraspecific and Interspecific Crossing"

_plants, 2025, doi:10.3390/plants14192985_

Round 1

Reviewer 1 Report

Comments and Suggestions for Authors

The research presented in this manuscript evaluated the possibility to evaluate the leaf coffe size and characteristics based on a non destructive methode.

The Abstract is a good presention of the work performed, presenting the main results.

The introduction provides a good litterature review with a lot of interesting bibliographic references which support the research question presented at the end of the introduction.

The results section is well explained and supported by the illustration which are of quality, I've missed a little a first table or figure and an associated paragraph as starting point of the results to sum up the global characteristics of the genotypes tested. I've seen the supplemental material, but I think you can consider introducing a short table to explain the differences encountered withing a genotype, and in the intra a inter crossed. Because one of the questions I'd is that you have enough variability within the progeny to be able to evaluate the performance of your models. You explain it more in the discussion, but it's worth value to have a resume idea before seen the results of the two models. And could be useful also to see a figure with two-three 'types' of leaves.

Then the discussion is complete again with abundant litterature.

The material and method section is complete and reproductible.

The conclusions are on line with the results obtained

Two minor form corrections, in the text references should be a number, not the names.

In line 347 there is a espace missing between statistical and significant.

Author Response

Reviewer 1.

Comment 1:

The research presented in this manuscript evaluated the possibility to evaluate the leaf coffe size and characteristics based on a non-destructive method.

Response 1:

We agree with this comment.

Comment 2:

The Abstract is a good presentation of the work performed, presenting the main results.

Response 2:

We agree with this comment.

Comment 3:

The introduction provides a good literature review with a lot of interesting bibliographic references which support the research question presented at the end of the introduction.

Response 3:

We agree with this comment.

Comment 4:

The results section is well explained and supported by the illustration which are of quality, I've missed a little a first table or figure and an associated paragraph as starting point of the results to sum up the global characteristics of the genotypes tested. I've seen the supplemental material, but I think you can consider introducing a short table to explain the differences encountered withing a genotype, and in the intra a inter crossed. Because one of the questions I'd is that you have enough variability within the progeny to be able to evaluate the performance of your models. You explain it more in the discussion, but it's worth value to have a resume idea before seen the results of the two models. And could be useful also to see a figure with two-three 'types' of leaves.

Response 4:

When referring to intraspecific and interspecific crosses, the terms allude to the collective group of all genotypes derived from these respective crosses and to the generalized calculation encompassing all genotypes. This is detailed in the materials and methods table. For this reason, we did not consider the introduction of an additional table necessary.

However, we have spelled out the full names of the variables throughout the abstract upon first use before introducing their abbreviations.

Regarding a figure with the leaf types, it would not be particularly informative because, while differences in the parameters do exist, they are very slight. As detailed in the discussion (Line 334), these minor variations could be attributable to mean asymmetry, where one side of the leaf is larger than the other.

Comment 5:

Then the discussion is complete again with abundant litterature.

Response 5:

We agree with this comment.

Comment 6:

The material and method section is complete and reproductible.

Response 6:

We agree with this comment.

Comment 7:

The conclusions are on line with the results obtained

Response 7:

We agree with this comment.

Comment 8:

Two minor form corrections, in the text references should be a number, not the names.

Response 8:

It was corrected in the new paper version.

Comment 9:

In line 347 there is a space missing between statistical and significant.

Response 9:

It was corrected in the new paper version.

Reviewer 2 Report

Comments and Suggestions for Authors

Well written, methods and results are clear.

Author Response

Reviewer 2

Comment 1:

Well written, methods and results are clear.

Response 1:

We agree with this comment.

Reviewer 3 Report

Comments and Suggestions for Authors

This study examines the applicability of the Montgomery model and the similarity principle model for non-destructive estimation of leaf area in Coffea arabica. Parameter analysis and cluster comparisons were conducted using data from 55 intra- and inter-specific hybrid progeny. While the research topic holds practical value, the paper exhibits shortcomings in methodological innovation, depth of data interpretation, and the accuracy of the “non-destructive” concept. The following revision suggestions are proposed:

  1. Insufficient methodological innovation

The study primarily relies on existing Montgomery and similarity principle models for data fitting and validation, lacking substantive methodological innovation. Authors should more clearly articulate the novelty of this research, such as why systematically comparing α and β parameters across 55 hybrid progeny represents a breakthrough. Does this comparison reveal how different genetic backgrounds influence model parameters for leaf area estimation? Does it imply that different models or parameters should be selected for breeding applications based on population differences? These aspects warrant further discussion to highlight the study's innovation and practical value. Additionally, consider integrating computer vision or deep learning methods to enhance methodological cutting-edge potential.

  1. Data interpretation andsStatistical methods require greater rigour

Supplementary data indicate pronounced leaf trait variations among hybrid progeny, yet the main text lacks sufficient explanation of the genetic and physiological mechanisms underlying these differences. The statistical analysis section (e.g., α estimation method, k value selection for clustering, multiple comparison adjustments for hypothesis testing) also lacks transparency, compromising reproducibility. Recommendations: ① Provide detailed steps for parameter estimation and statistical testing in the Methods section; ② Strengthen biological interpretations of intra- and inter-hybrid population differences in Results and Discussion; ③ Specify the applicability and limitations of α≈0.6700.

  1. The term “non-destructive” requires clarification

While the article title and abstract emphasize “non-destructive,” the method still requires removing leaves for length and width measurements. This differs from truly in situ non-destructive techniques (e.g., image recognition, sensor measurements). The authors are advised to adopt more precise terminology, such as “indirect estimation method based on leaf length and width,” and to address the differences and limitations compared to genuinely non-destructive techniques in the Discussion section.

  1. Presentation of results and figure readability require improvement

Current tables (e.g., Table 1, Table 2) contain excessive data, with verbose repetition in the text making it difficult for readers to grasp key points. The paper primarily relies on large tables, lacking intuitive images or graphical representations. The absence of photographs illustrating typical coffee leaf morphology hinders readers' understanding of the biological significance of metrics like leaf width/length ratios and ellipticity indices. Recommendations: ① Highlight key findings in the main text (e.g., differences in α parameters between inter- and intra-hybrid populations); ② Add significance annotations or grouped visualizations to figures/tables; ③ Include relevant images in the main text or supplementary materials, such as collection site photos, typical leaf images, and measurement diagrams.

  1. Application value requires further expansion

The paper emphasizes the method's convenience but lacks depth in discussing practical application value. Recommendations: ① Outline specific advantages and economic benefits of this method in coffee breeding and field monitoring; ② Explore future integration with technologies like image recognition and remote sensing to enhance the paper's forward-looking perspective and practical utility.

Author Response

Reviewer 3

Comment 1:

This study examines the applicability of the Montgomery model and the similarity principle model for non-destructive estimation of leaf area in Coffea arabica. Parameter analysis and cluster comparisons were conducted using data from 55 intra- and inter-specific hybrid progeny. While the research topic holds practical value, the paper exhibits shortcomings in methodological innovation, depth of data interpretation, and the accuracy of the “non-destructive” concept. The following revision suggestions are proposed:

Insufficient methodological innovation

The study primarily relies on existing Montgomery and similarity principle models for data fitting and validation, lacking substantive methodological innovation. Authors should more clearly articulate the novelty of this research, such as why systematically comparing α and β parameters across 55 hybrid progeny represents a breakthrough. Does this comparison reveal how different genetic backgrounds influence model parameters for leaf area estimation?

Response 1:

The scope of this study is not intended to reveal how genetic background influences the parameters. It is limited to reporting that the progenies derived from interspecific hybrids (Inter.H.) and classified in cluster 2 exhibit characteristics linked to C. canephora, as would be expected. However, given the nature of the crosses and the degree of arabization, these traits are not exclusive to the interspecific hybrids. A paragraph has been added to provide this explanation (line 382).

Comment 2:

Does it imply that different models or parameters should be selected for breeding applications based on population differences?

Response 2:

There are no significant differences between the populations derived from intraspecific (Intra.H) and interspecific (Inter.H) hybrids, as detailed in Table 1 and at the beginning of the Discussion section. Hence, we propose the option of using a general value when evaluating leaf size in composite varieties (which consist of multiple progenies). However, in the case of single-progeny varieties, the variability of the α parameter suggests that it should be estimated if prior information is not available. This is detailed in the Discussion (lines 306-308).

Comment 3:

These aspects warrant further discussion to highlight the study's innovation and practical value. Additionally, consider integrating computer vision or deep learning methods to enhance methodological cutting-edge potential.

Response 3:

The integration of computer vision and deep learning methods, while interesting, is beyond the scope of the present study. Furthermore, given the "black box" and complex nature of how weights are calculated in such models, they do not yield a result that is easily comparable across other species. This is in contrast to the α parameter of the Montgomery model. While not necessarily the best method, the α parameter is widely used across different species and provides an easily comparable result.

Comment 4:

Data interpretation and Statistical methods require greater rigour

Supplementary data indicate pronounced leaf trait variations among hybrid progeny, yet the main text lacks sufficient explanation of the genetic and physiological mechanisms underlying these differences. The statistical analysis section (e.g., α estimation method, k value selection for clustering, multiple comparison adjustments for hypothesis testing) also lacks transparency, compromising reproducibility. Recommendations:

① Provide detailed steps for parameter estimation and statistical testing in the Methods section;

Response 4:

The parameter estimation is detailed in the Materials and Methods section. The process consists of performing a non-linear regression, with the distinction that the parameters are calculated using bootstrapping. This is carried out automatically using the “minpack.lm” package. The authors do not consider a further explanation of this point to be necessary.

Comment 5:

② Strengthen biological interpretations of intra- and inter-hybrid population differences in Results and Discussion;

Response 5:

The biological interpretations are presented in the Results and Discussion sections. However, they are limited to the documented foundations for the α and β parameters, as the differences between the populations derived from intraspecific (Intra.H) and interspecific (Inter.H) hybrids have not been thoroughly documented in other studies.

Comment 6:

③ Specify the applicability and limitations of α≈0.6700.

Response 6:

The document details that the α value of ≈0.6700 should be limited to groups of progenies, such as composite varieties.

Comment 7:

The term “non-destructive” requires clarification

While the article title and abstract emphasize “non-destructive,” the method still requires removing leaves for length and width measurements. This differs from truly in situ non-destructive techniques (e.g., image recognition, sensor measurements). The authors are advised to adopt more precise terminology, such as “indirect estimation method based on leaf length and width,” and to address the differences and limitations compared to genuinely non-destructive techniques in the Discussion section.

Response 7:

The method does not require the removal of leaves. It should be noted that an initial, destructive sampling of leaves is necessary to calculate the actual leaf area and its corresponding length and width measurements. Once this relationship is established, one can proceed with estimating the α parameter for the Montgomery model. Once this parameter has been estimated (as done in the present article), only a ruler is needed to measure the dimensions of any new leaf and apply them to the model, from which an estimated leaf area can be calculated. In this sense, the subsequent application of the model is non-destructive, a fact supported by the associated scientific literature on this topic.

Comment 8:

Presentation of results and figure readability require improvement

Current tables (e.g., Table 1, Table 2) contain excessive data, with verbose repetition in the text making it difficult for readers to grasp key points. The paper primarily relies on large tables, lacking intuitive images or graphical representations. The absence of photographs illustrating typical coffee leaf morphology hinders readers' understanding of the biological significance of metrics like leaf width/length ratios and ellipticity indices. Recommendations:

① Highlight key findings in the main text (e.g., differences in α parameters between inter- and intra-hybrid populations);

Response 8:

In the document, it is established that there are no differences between the populations derived from Intra.H and Inter.H.

Comment 9:

② Add significance annotations or grouped visualizations to figures/tables;

Response 9:

Given the amount of information, and based on the presentation format found in other high-impact publications (e.g., Yu, X., P. Shi, J. Schrader, and K. J. Niklas. 2020. Nondestructive estimation of leaf area for 15 species of vines with different leaf shapes. American Journal of Botany. 107(11): 1481–1490), the current data presentation format has been retained.

Comment 10:

③ Include relevant images in the main text or supplementary materials, such as collection site photos, typical leaf images, and measurement diagrams.

Response 10:

The differences among coffee leaves are very subtle, as the Sh parameter reveals; however, they are statistically significant. Furthermore, these differences may arise from asymmetry between the sides of the leaf—an aspect that was observed but not evaluated in the present study. For these reasons, photographs are not included.

Comment 11:

Application value requires further expansion

The paper emphasizes the method's convenience but lacks depth in discussing practical application value. Recommendations:

① Outline specific advantages and economic benefits of this method in coffee breeding and field monitoring;

Response 11:

This point is clearly detailed in the introduction (lines 49–65).

Comment 12:

② Explore future integration with technologies like image recognition and remote sensing to enhance the paper's forward-looking perspective and practical utility.

Response 12:

This falls outside the scope of the manuscript, as it corresponds to a more classical approach, albeit one that estimated the parameters using current statistical tools.

Round 2

Reviewer 3 Report

Comments and Suggestions for Authors

The study employs the Montgomery model and the similarity principle model to estimate leaf area in 55 Arabica coffee hybrids and provides a systematic comparison of parameters, particularly the applicability and limitations of the α coefficient. While the work has certain practical relevance, it lacks sufficient originality, as it primarily applies existing models for fitting and validation without introducing novel methodological approaches or clear conceptual advances.

Moreover, the depth of discussion and interpretation is limited. The biological mechanisms underlying the observed differences among hybrid populations are only superficially addressed, without deeper insights into genetic or physiological factors. The discussion of practical value remains rather general and does not provide sufficient analysis of the implications for breeding or field management. In addition, the presentation of results is not fully effective: the manuscript relies heavily on large tables and lacks intuitive visual illustrations, while the forward-looking perspective, particularly connections to more advanced non-destructive approaches, is underdeveloped.

In summary, the manuscript appears more as an application-oriented validation of established methods on new plant material rather than a substantive advancement in theory or methodology. For these reasons, it does not meet the level of innovation and depth expected for publication in Plants. I would therefore recommend rejection, while encouraging the authors to strengthen the novelty, depth, and forward-looking aspects of the study.

Author Response

Comment:
The study employs the Montgomery model and the similarity principle model to estimate leaf area in 55 Arabica coffee hybrids and provides a systematic comparison of parameters, particularly the applicability and limitations of the α coefficient. While the work has certain practical relevance, it lacks sufficient originality, as it primarily applies existing models for fitting and validation without introducing novel methodological approaches or clear conceptual advances.

Response:

The authors understand the reviewer's comment. However, the study's objective was to find apparent differences between the progenies from intraspecific (Intra.H) and interspecific (Inter.H) hybrids, which was not the case. Instead, it was found that progenies in both groups can exhibit a common leaf type, a finding that was established by the cluster analysis. A paragraph has been added to state this explicitly (lines 402–406). Similarly, our findings allow us to propose that a value of 0.67 is an appropriate general α parameter for Coffea or for groups of progenies. The direct comparison between Intra.H. and Inter.H. lines has not been documented in any previous work; hence, we consider our study to be both relevant and novel.

Comment:

Moreover, the depth of discussion and interpretation is limited. The biological mechanisms underlying the observed differences among hybrid populations are only superficially addressed, without deeper insights into genetic or physiological factors. The discussion of practical value remains rather general and does not provide sufficient analysis of the implications for breeding or field management. In addition, the presentation of results is not fully effective: the manuscript relies heavily on large tables and lacks intuitive visual illustrations, while the forward-looking perspective, particularly connections to more advanced non-destructive approaches, is underdeveloped.

Response:

We thank you for the comment. The manuscript states that there are no significant differences between the Intra.H and Inter.H groups as a whole. Rather, what exists are progenies that may exhibit a particular shape, but this shape is not exclusive to either type of hybridization. Therefore, an explanation of biological mechanisms should not be directed toward the progenies themselves. We have added a sentence suggesting that the lack of differences between Intra.H and Inter.H is due to the high degree of arabization, as indicated by the generation of the cross (lines 348–351). Furthermore, understanding the underlying biological mechanisms would require additional information, for example, data on shading factors and the light extinction coefficient to elucidate how leaves alter their size in response to these conditions or, alternatively, the use of contrasting localities. This is clearly beyond the scope of the present investigation and the available data.

Following the reviewer's recommendation, we have added a paragraph that denotes the possible implications for genetic improvement (lines 386–389).

We understand that the presentation of the results may not be the most visually appealing. However, we consider it relevant for the reader to see the parameter's confidence limits, given the non-symmetrical nature of the limits calculated by the Bootstrap method around the mean. Additionally, given the high number of progenies, this is a challenge that is difficult to overcome even with a purely visual representation.

Comment:

In summary, the manuscript appears more as an application-oriented validation of established methods on new plant material rather than a substantive advancement in theory or methodology. For these reasons, it does not meet the level of innovation and depth expected for publication in Plants. I would therefore recommend rejection, while encouraging the authors to strengthen the novelty, depth, and forward-looking aspects of the study.

Response:

We appreciate the time spent on our manuscript.

Round 3

Reviewer 3 Report

Comments and Suggestions for Authors

I am fine with the manuscript being published.